# Study of the Effects of Physical-Activity Practice and Adherence to the Mediterranean Diet on Emotional Intelligence in Elementary School Education Students

**DOI:** 10.3390/children10071211

**Published:** 2023-07-12

**Authors:** Eduardo Melguizo-Ibáñez, José Luis Ubago-Jiménez, Gabriel González-Valero, Georgian Badicu, Sameer Badri Al-Mhanna, Pilar Puertas-Molero

**Affiliations:** 1Faculty of Education Sciences, Department of Didactics of Musical, Plastic and Corporal Expression, University of Granada, 18071 Granada, Spain; emelguizo@ugr.es (E.M.-I.); jlubago@ugr.es (J.L.U.-J.); ggvalero@ugr.es (G.G.-V.); pilarpuertasmolero@gmail.com (P.P.-M.); 2Department of Physical Education and Special Motricity, Faculty of Physical Education and Mountain Sports, Transilvania University of Braşov, 500068 Braşov, Romania; 3Department of Physiology, School of Medical Sciences, Universiti Sains Malaysia, Kubang Kerian 16150, Kelantan, Malaysia; sameerbadri9@gmail.com

**Keywords:** elementary education, physical activity, Mediterranean diet, emotional intelligence

## Abstract

Research shows that regular physical exercise and a healthy lifestyle have a beneficial effect on the health of adolescents. Other studies have also shown that gender is also a determining factor when it comes to leading an active and healthy lifestyle. The aims of this study are (a) to develop a structural-equation model formed by the variables of emotional intelligence, physical-activity practice, and adherence to the Mediterranean diet and (b) to consider that model through a multigroup analysis. A descriptive, comparative, and cross-sectional study was carried out. The sample consisted of 567 primary school students. The KIDMED, PAQ-C, and TMMS-24 questionnaires were used for data collection. We found that there are differences in the effects of the practice of physical activity and adherence to the Mediterranean diet on the emotional domain. We also concluded that, during adolescence, gender is a key element in acquiring a healthy and active lifestyle.

## 1. Introduction

The concept of “adolescence” is very useful in discussing the different stages of human development. Specifically, this stage of development is characterized by many physical, conceptual, and emotional changes in an individual [1]. Likewise, the molding of the personality and the acquisition of healthy habits also take place [2]. Although this stage of development has a high degree of relevance in terms of the acquisition of a healthy lifestyle, it has been observed that gender also affects adherence to a healthy and active lifestyle [3]. Gender differences are related to maturational development, as during adolescence the female population tends to opt for more sedentary activities [4]. In contrast, male teenagers tend to show an increase in weekly-physical-activity time [4]. Specifically, it has been observed that during adolescence, female individuals show a detachment from physical-activity practice [3]. It has also been noted that during adolescence, both young males and females show an excessive intake of hypercaloric foods [5], which is detrimental to health [5].

With the new dietary changes that are taking place, Papadaki et al. [6] affirmed that the intake of hypercaloric foods depends on the lifestyles of families [6]. It has been observed that the inclusion of both parents in the labor market leads to less time devoted to cooking food [7], resulting in less concern for healthy food intake [7]. Easy access to the intake of hypercaloric foods is also promoted by food delivery companies, where adolescents have greater access to precooked foods with a high caloric level [8]. 

Notwithstanding the current problem, research by Muros et al. [9] affirmed that the Mediterranean diet is a healthy dietary model that has numerous benefits for the health of young people. Specifically, this model, which is low in animal fats and refined sugars, is characterized by a predominance of fresh products [10], including vegetables, fruits, cereals, and legumes and a high consumption of oily fish, eggs, dairy products, and omega-3 fatty acids [10]. Although the Mediterranean diet has different variations, two main characteristics are found: macronutrient intake (between 30–35% fat, 53–58% carbohydrates, and 10–12% protein) and fat quality (between 7–10% saturated fatty acids, 15–20% monounsaturated fatty acids, and 6–8% polyunsaturated fatty acids) [11]. The scientific literature establishes that positive adherence to the Mediterranean diet has a beneficial impact on the health of individuals at the physical, mental, and emotional levels [9,12]. It has also been pointed out that a healthy lifestyle, together with an active lifestyle, helps to increase life expectancy [13].

In this context, physical-activity practice is defined as the movement originated by any skeletal muscle derived from a muscular contraction that originates a significant expenditure of energy in a person [14]. Currently, the scientific literature indicates that adolescents show higher degrees of sedentary lifestyles [15]. This is mainly due to technological developments, as young people prefer to carry out sedentary activities [16] that do not result in high energy expenditure. In order for the practice of physical activity to be beneficial for the body, criteria such as the frequency of exercise, the duration of exercise, exercise intensity, and the type of sports practice should be taken into account [17]. Together with these criteria, the World Health Organization [18] has established that, for the age range between 5 and 17 years, at least 60 min of intense and moderate physical activities per day should be carried out, and that such activities should be aerobic and game-oriented. It has been observed that the practice of regular physical activity has a beneficial impact on individuals’ mental level, as it leads to an improvement in physical appearance, resulting in an increase in self-concept and self-esteem [19].

New lines of study on the benefits of physical activity have determined that the regular practice of physical exercise helps to improve people’s emotional states [20]. One of the key elements of emotionally competence is emotional intelligence [21]. It has been observed that emotional intelligence is a factor that directly influences people’s moods [20]. That is why emotional intelligence is defined as a set of skills that allow people to assimilate, perceive, and regulate both their own emotions and those of people around them [21]. 

Emotional intelligence is composed of three dimensions [21]. The first area is emotional attention [22], which is the ability to understand what causes emotions in one’s self and in others [22]. Then, emotional clarity is considered [21]; it consists of the ability to discern whether an emotion acts negatively or positively on a person [22]. Finally, there is emotional repair [22], which consists of overcoming emotions that have a negative effect on a person [22]. It has been shown that the correct use of emotions has a great impact on the executive functions of young people [23]. It has been observed that emotional skills promote mental processes and help to improve executive functions [23], especially concentration in times of stress and anxiety [23].

Within the Spanish education system, there is increasing pressure on young people to perform better academically [5,9,10]. As a result of this pressure, many young people may experience negative emotional states, such as anxiety and stress [24]. Continued subjection to these negative emotions can act negatively on academic performance and other areas of human performance [25]. Therefore, young people should be educated to be physically active in physical education classrooms [24]. It has been shown that regular physical exercise helps to decrease the levels of negative emotions [24]. 

This research brings to the scientific field a study of the effects of physical exercise on the three variables that make up emotional intelligence. The effects of these variables on emotional intelligence are different, and each are studied individually. These effects are also studied according to the sex of the participants, which allows for a more complete study of the differences between the male and female population. 

The following research hypotheses were considered: 

**H1.** *Mediterranean-diet adherence and physical-activity practice have a positive effect on emotional intelligence*. 

**H2.** 
*There are differences, according to gender, in the effects of the practice of physical activity.*


Finally, this research aims to (a) establish a structural-equation model formed by the variables of Mediterranean-diet adherence, physical-activity practice, and emotional intelligence and (b) study the differences in the effects of the variables through a multigroup model.

## 2. Materials and Methods

### 2.1. Design and Participants

This research follows a non-experimental (ex post facto) cross-sectional and descriptive design. A random criterion was used to select the different educational centers where the data were to be collected. The sample under study consisted of students in the primary education stage. Specifically, the sample consisted of students in the second and third cycles of this educational stage. The ages of the participants were between 9 and 12 (11.10 ± 1.24). A total of 567 elementary-school students participated in this research. Of the participants, 53% were male (*n* = 303) and 47% (*n* = 264) were female. The families of these participants belonged to the middle socio-economic level. To ensure that the sample was representative, a sampling-error study was applied: for a confidence level of 95%, a sampling error of less than 5% was obtained.

### 2.2. Instruments and Variables

**Self-prepared questionnaire:** This self-developed instrument was used to collect socio-demographic data, such as data regarding gender (male/female) and age. 

**Trait Meta Mood Scale [26]**: This instrument was employed to collect data related to emotional intelligence. Although the original version [26] was developed in English, the Spanish version [27] was used for this study. It consisted of 24 items that were answered, based on a Likert scale. This instrument, which was developed under the trifactorial emotional intelligence model [20], evaluated this variable of emotional intelligence from a three-dimensional perspective. The first eight items of collected data were focused on emotional attention (EA). The second eight items of collected data were focused on emotional clarity (EC). The remaining items of collected data focused on emotional repair (ER). To ensure that the data were reliable, Cronbach’s alpha test was used. For this research, the following values were obtained: EA α = 0.889; EC α = 0.824; ER α = 0.845.

**KIDMED Questionnaire [28]**: This instrument was applied to collect data related to Mediterranean-diet adherence. It consisted of 16 questions that were answered in a positive or negative way. Specifically, questions 6, 12, 14, and 16 were negatively worded. If they were answered positively, this was interpreted as −1 point. The Cronbach’s alpha test obtained a value of α = 0.742.

**PAQ-C questionnaire:** This questionnaire was employed to collect data related to the practice of physical activity. Specifically, the Spanish version was used for this study [29]. It consisted of 10 questions that evaluated the type and frequency of physical activity carried out in the last seven days. The Cronbach’s alpha test for this instrument obtained a value of α = 0.746.

### 2.3. Procedure

Before starting the fieldwork, several review studies were carried out. The purpose of these studies was to determine the most reliable instruments for data collection. Different research studies were considered to determine the best possible research design.

Because the present study was carried out within a small population, an informative letter was initially sent to educational centers, informing them of the purpose and objectives of the study. Once permission to access the different educational centers was obtained, an informative letter was sent to the legal guardians of the young people who were involved. This letter informed the legal guardians of the type of study and its objectives. Once the legal guardians had given their informed consent, the questionnaires previously mentioned could be administered. 

Data collection was carried out during the physical education hour. The physical education teacher and the research team were present during data collection. The research team consisted of PhD specialists in physical education, physical activity, and sport sciences. During data collection, the research team was present to resolve any doubts that arose. The data were treated anonymously, and were used exclusively for scientific purposes. The research was conducted between January 2023 and May 2023. Specifically, the research team had access to three schools. The first of these schools was accessed in January 2023. The second school was accessed in March 2023. The third school was accessed in May 2023. The study followed the ethical principles set out in the Declaration of Helsinki. It was also supervised by an ethics committee of the University of Granada (2966/CEIH/2022).

### 2.4. Data Analysis

Given the type of analysis that was under consideration, it was not necessary to study the normality of the data. Following the approach of Hair et al. [30], it was necessary to take into account (a) the sample size, (b) the minimum ratio of cases per variable, and (c) the ratio of variables per factor. The IBM SPSS Amos 26.0 (IBM Corp., Armonk, NY, USA) program was employed to develop the three structural-equation models for this research. Each of the models consisted of five variables. According to their characteristics, three of the variables acted as endogenous variables and two of the variables acted as exogenous variables (Figure 1). Due to the characteristics of the endogenous variables, causal relationships were introduced, because of the associations observed between the degree of reliability and the measurement indicators. The inclusion of causal relationships allowed the inclusion in the three models of the errors derived from the measurement process of the observed variables. For the endogenous variables, the error derived from the measurement process (e1, e2, e3) was added.

For the elaboration and interpretation of the structural-equation models, the direction of the arrows is considered (Figure 1). The unidirectional directions should be interpreted from the regression weights. In addition, these directions can be interpreted as lines of influence. The directions symbolize the directions of the effects. Finally, the degrees of significance were set at *p* ≤ 0.05 and *p* ≤ 0.001.

To evaluate the fit of a model, the recommendations proposed by Maydeu-Olivares [31] and Kyriazos [32] were followed. They [31,32] stated that, in order to satisfactorily fit a model, different indices should be taken into account, including the comparative fit index (CFI), the incremental reliability index (IFI), the goodness-of-fit Index (GFI), and the root-mean-square approximation (RMSEA). Although these indices are sufficient to satisfactorily fit a structural model, the statistical sensitivity of the sample size [32] has also been considered, so the Tucker–Lewis index (TLI) was also introduced. Scores above 0.90 for the CFI, IFI, GFI, and TLI showed a good fit [31,32,33]; however, for the RMSEA, the values had to be below 0.100 [31,32,33]. Table 1 shows the fit indices of the CFI, IFI, GFI, and TLI values for the three structural-equation models.

## 3. Results

The proposed model for the sample obtained a good fit in the different indices. The chi-square test (X^2^ = 19.971; df = 13; pl = 0.213) obtained a non-significant value, indicating a good fit [30]. Attending to the values of the different fit indices, values higher than 0.900 were obtained for the CFI, IFI, GFI, and TLI. For the RMSEA, a value of 0.084 was obtained. The values obtained in a model’s fit indices show a good fit for each of the models.

Table 2 and Figure 2 show the results for the effects of the proposed model. A positive effect of adherence to the Mediterranean diet on emotional attention (EA) (β = 0.109), emotional clarity (EC) (*p* ≤ 0.05; β = 0.138), and emotional repair (ER) (*p* ≤ 0.05; β = 0.175) was observed. Similarly, a positive effect of physical activity on EA (β = 0.007), EC (β = 0.073), and ER (β = 0.079) was observed.

The structural models proposed regarding participants’ gender showed a good fit. The males’ model showed a non-significant value (X^2^ = 20.164; df = 12; pl = 0.001), and the same was true for the females’ model (X^2^ = 22.973; df = 9; pl = 0.000). Regarding the fit indices, values above 0.900 were obtained for the CFI, IFI, GFI, and TLI. The RMSEA showed a value of 0.079 for the model proposed for males and 0.093 for the model proposed by females. The values obtained for the fit indices of these models show a good fit.

Figure 3 and Figure 4, together with Table 3 and Table 4, show the results obtained according to the gender of the participants. For males, the effect of greater adherence to the Mediterranean diet on emotional clarity (*p* ≤ 0.05; β = 0.259) and emotional reappraisal (*p* ≤ 0.05; β = 0.197) was found. In contrast, for females, a greater effect of Mediterranean-diet adherence on emotional attention was found (β = 0.667). Regarding the effect of physical activity on the emotional area, a greater effect was observed for male teenagers (*p* ≤ 0.05 β = 0.187; *p* ≤ 0.05 β = 0.284; *p* ≤ 0.05 β = 0.151).

## 4. Discussion

Once the objectives and the initial hypotheses were identified, the aim of this study was to contextualize the findings obtained in this research.

The analysis of the study population showed a positive effect of adherence to the Mediterranean diet on the three areas that comprise emotional intelligence. The results were very similar to those obtained by Melguizo-Ibáñez et al. [34], who affirmed the numerous benefits of adherence to the Mediterranean diet on physical self-concept [35]. Assessment tests have also shown that many young people have a higher intake of high-fat foods [36]. In view of this, it has been found that young people who demonstrate emotional control showed control over these disruptive states, leading to healthy food intake [37]. Specifically, adolescents who palliate negative emotional states through uncontrolled food intake are more reluctant to change this type of behavior [38]. This is mainly due to the fact that during the process of overeating, uncontrolled food intake generates a positive effect [38].

A positive effect of the practice of activity on the emotional dimensions was observed. The research carried out by Li et al. [20] indicated that te physical-sports practice provides numerous benefits, as it produces a decrease in endorphins and eliminates norepinephrine and cortisol, two hormones that appear in situations of prolonged stress [39].

The multigroup analysis according to the declared gender of the participants showed a greater effect of adherence to the Mediterranean diet on the emotional construct of males. Tsochantaridou et al. [8] affirmed that young people in different Western societies showed greater care for their body image, which led to benefits at the emotional level. Regarding gender differences, research by Grams et al. [40] found that boys generally tend to show greater detachment from a healthy dietary pattern and, thus, a worse conception of their emotional states. In contrast, research by Ferrer-Cascales et al. [41] found a better conception of emotional and dietary states for females. Similarly, Melguizo-Ibáñez et al. [42] found that female teenagers showed greater emotional competence than males.

Regarding the effect of the practice of physical activity, a greater effect was found for the areas of emotional intelligence for males. It has been found that females during adolescence have a higher level of sedentary lifestyles and that such lifestyles have a negative influence on the emotional area [43]. In addition, a negative effect of the abandonment of regular physical activity on attention and emotional clarity was observed for females. These results are very different from those found by Wang et al. [44], who affirmed that the segregation of neurotransmitters helped to channel disruptive emotional states. Nevertheless, it can be affirmed physical-sports practice can condition the appearance of emotional states or eliminate negative emotional states [45]. The abandonment of the practice of physical activity by women may be due to the choice to carry out activities with a higher level of sedentary lifestyle [46].

Numerous studies have suggested that adolescent populations are becoming increasingly sedentary [47,48]. Numerous studies have affirmed that the educational field must provide a comprehensive education, in which the complete development of the student is achieved [34,42,49]. It has been found that, apart from the educational environment, there are other variables that can condition the acquisition of an active and healthy lifestyle. One of them is the family environment, as it has been found that a higher degree of family functionality helps to transmit a positive motivation toward physical-activity practice [6]. New research has suggested that the socioeconomic status of families promotes an active and healthy lifestyle [50]. It has also been shown that another element that influences an active and healthy lifestyle is motivation [51]. During adolescence there is a decrease in the time of physical activities, especially in girls [51]. The interests toward which the subject of physical education is oriented can have a positive or negative impact on the motivations of students [52,53].

It has been shown that physical inactivity can have a negative impact on the physical and mental capacities of adolescents [54]. If habits of regular physical activity are not created during childhood and adolescence, a higher level of a sedentary lifestyle is likely during adulthood [55]. To develop a habit related to the practice of physical activity, numerous studies have highlighted the role of the physical education teacher [56]. It has been observed that teachers who orient physical-activity practice toward enjoyment and fun manage to create a positive bond with their students [57]. It has also been shown that a physical education pedagogy focused on enjoyment helps students to reduce anxiety and stress levels and to improve their emotional states [58].

The strengths and limitations of this research should be mentioned. In terms of strengths, this study provides a multi-group analysis based on the gender of adolescents. This provides a step forward in the analysis of differences in the physical-health domain. In addition, from these data, an intervention program can be developed that focuses on the gender preferences of students. This type of analysis has not been widely carried out in terms of considering the differences between genders—thus, a new study typology is proposed. 

Regarding the limitations of this study, it should be mentioned that the research design was a limitation. Because only one data collection was carried out, the cause–effect relationships of these variables were studied only at that time point. As for the questionnaires, although they had a high degree of reliability, they showed an intrinsic measurement error. In addition, the sample belonged to a very specific geographic area, so no generalizations could be made. It would have been interesting to measure the socioeconomic levels of the families, as well as the degrees of family functionality.

Finally, new lines of research or projects may emerge from this study. The first of these would be an experimental study to determine the motivations of participants for abandoning regular physical activity as a function of the gender. It would also be interesting to carry out an exploratory study after an intervention program has been implemented. Through these types of analysis, it would be possible to examine the variations in the effects after an intervention program. It would also be interesting to include a study of new variables, such as the socioeconomic levels of families.

## 5. Conclusions

The present study shows that in the elementary education stage, students in the third cycle receive positive effects from both Mediterranean-diet adherence and physical activity on the emotional domain. 

In considering the multigroup structural-equation models in terms of gender, differences were observed between the effects of the Mediterranean diet and physical activity on emotional intelligence. For male teenagers, a greater effect of Mediterranean-diet adherence and physical activity on emotional intelligence was observed. In this regard, the two research hypotheses initially established were fulfilled.

Finally, this study highlights the importance of studying gender differences in the adolescent population when it comes to leading an active and healthy lifestyle. To this end, physical education teachers must know the previous interests of their students and orient their physical education classes according to these interests. To achieve this objective, teachers must take into account the motivational climate toward which his or her physical education classes are oriented.

## Figures and Tables

**Figure 1 children-10-01211-f001:**
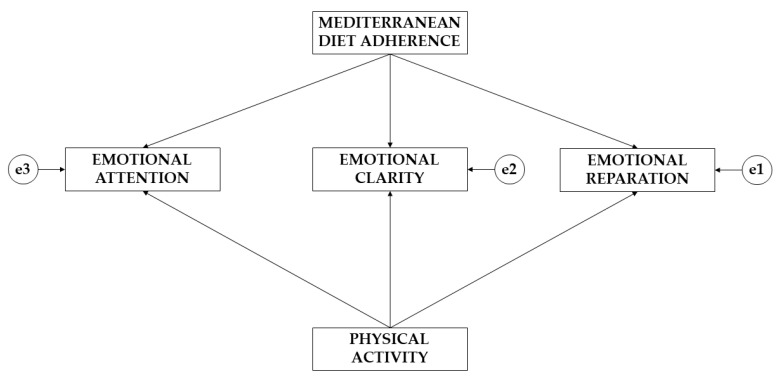
Theoretical model.

**Figure 2 children-10-01211-f002:**
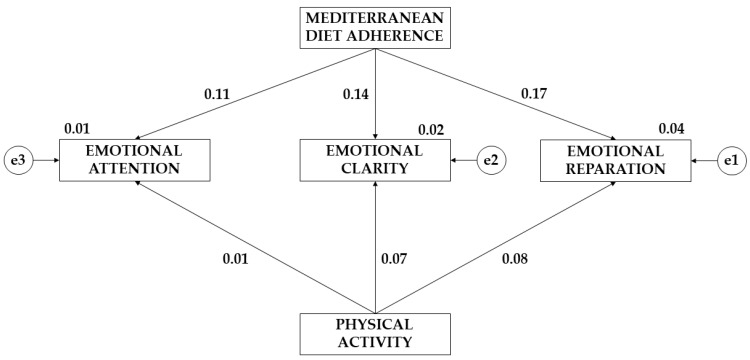
Theoretical model with the results of the effects for the whole sample.

**Figure 3 children-10-01211-f003:**
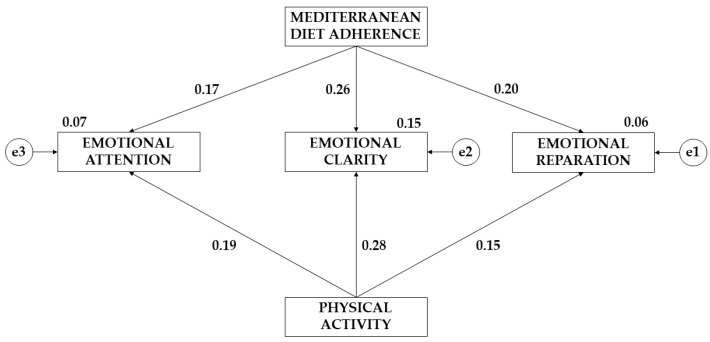
Theoretical model with the results of the effects for male teenagers.

**Figure 4 children-10-01211-f004:**
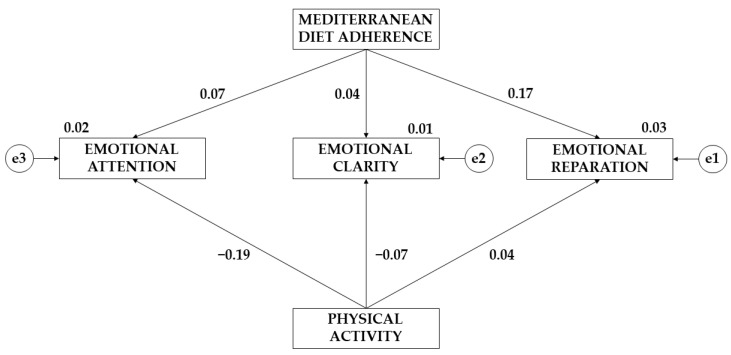
Theoretical model with the results of the effects for female teenagers.

**Table 1 children-10-01211-t001:** Values of the adjustment indices.

	CFI	IFI	GFI	TLI
**Entire Sample Model**	0.972	0.978	0.963	0.949
**Male Model**	0.968	0.970	0.944	0.909
**Female Model**	0.990	0.995	0.927	0.900

**Table 2 children-10-01211-t002:** Effects obtained for the sample.

Effect Direction	Regression Weights				Standardised Regression Weights
	Estimations	Error Estimation	Critical Ratio	*p*	Estimations
EA←MDA	0.612	0.326	1.879	0.060	0.109
EC←MDA	0.651	0.272	2.392	0.017	0.138
ER←MDA	0.803	0.264	3.039	0.002	0.175
EA←PA	0.017	0.147	0.114	0.909	0.007
EC←PA	0.155	0.123	1.267	0.205	0.073
ER←PA	0.164	0.119	1.377	0.168	0.079

Note: emotional attention (EA); emotional clarity (EC); emotional reparation (ER); physical activity (PA); Mediterranean-diet adherence (MDA).

**Table 3 children-10-01211-t003:** Effects obtained for male participants.

Effect Direction	Regression Weights				Standardised Regression Weights
	Estimations	Error Estimation	Critical Ratio	*p*	Estimations
EA←MDA	0.891	0.411	2.171	0.030	0.174
EC←MDA	1.107	0.326	3.392	0.001	0.259
ER←MDA	0.850	0.347	2.451	0.014	0.197
EA←PA	0.516	0.221	2.332	0.020	0.187
EC←PA	0.654	0.176	3.722	0.001	0.284
ER←PA	0.352	0.187	1.883	0.060	0.151

Note: emotional attention (EA); emotional clarity (EC); emotional reparation (ER); physical activity (PA); Mediterranean-iet dherence (MDA).

**Table 4 children-10-01211-t004:** Effects obtained for female participants.

Effect Direction	Regression Weights				Standardised Regression Weights
	Estimations	Error Estimation	Critical Ratio	*p*	Estimations
EA←MDA	0.415	0.507	0.818	0.413	0.667
EC←MDA	0.198	0.433	0.457	0.648	0.038
ER←MDA	0.823	0.403	2.043	0.041	0.167
EA←PA	−0.278	0.195	−1.423	0.155	−0.117
EC←PA	−0.133	0.167	−0.797	0.425	−0.066
ER←PA	0.068	0.155	0.439	0.661	0.036

Note: emotional attention (EA); emotional clarity (EC); emotional reparation (ER); physical activity (PA); Mediterranean-aiet Adherence (MDA).

## Data Availability

The data used to support the findings of the current study are available from the corresponding author upon request.

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
