# Peer review of "Study of the Effects of Physical-Activity Practice and Adherence to the Mediterranean Diet on Emotional Intelligence in Elementary School Education Students"

_children, 2023, doi:10.3390/children10071211_

Round 1
Reviewer 1 Report
In line 75 they describe that the regular practice of physical exercise helps to improve people's emotional state. In the next line they talk about emotional intelligence and then give a definition of what it is about. In line 80 they write about people's self-image. The question that arises to me is that emotional intelligence is only the same as self-image? Or is it the same as emotional state? Or are both concepts included within emotional intelligence? I think this point needs to be clarified, since emotional intelligence is part of your goal.
In the paragraph that begins on line 84, they talk about the pressure of young Spaniards to achieve higher academic performance, the information is well supported, but I think that it is not related to what was stated above, since at no time do they state that the pressure academic affects any of its study variables. I think it is necessary to reconsider whether it is necessary to remove this paragraph, or to improve the wording.
In the version that I am reviewing there are several parts highlighted in yellow, this is the result of you uploading it like this by mistake, or when I download the file I see it like this.
When reading the methodology section, specifically the instruments, I see that within emotional intelligence there are other concepts. I think it is necessary to improve the wording of that part in the introduction so that the reader can easily understand the information.
In lines 257-258 they expose numerous investigations, but there is only one investigation that supports it. Improve writing or add more references.
Within the limitations of the study, it may be that they have used questionnaires to measure the variables, which may cause that ...
In the final part of the discussion, add information that highlights what is new about your research, and the practical utility of your results.
In the conclusions it is necessary to say which of the hypotheses were accepted or rejected.
Dear, they did an interesting job, it is only necessary to make small adjustments. I hope I have helped improve your research.
Greetings.
Author Response
Dear reviewer,
Comment 1
In line 75 they describe that the regular practice of physical exercise helps to improve people's emotional state. In the next line they talk about emotional intelligence and then give a definition of what it is about. In line 80 they write about people's self-image. The question that arises to me is that emotional intelligence is only the same as self-image? Or is it the same as emotional state? Or are both concepts included within emotional intelligence? I think this point needs to be clarified, since emotional intelligence is part of your goal.
Response 1
Thank you very much for your suggestion.
The paragraph you quote has been deleted. It has been replaced by the contextualisation of the variables attention, clarity and emotional repair.
Comment 2
In the paragraph that begins on line 84, they talk about the pressure of young Spaniards to achieve higher academic performance, the information is well supported, but I think that it is not related to what was stated above, since at no time do they state that the pressure academic affects any of its study variables. I think it is necessary to reconsider whether it is necessary to remove this paragraph, or to improve the wording.
Response 2
Thank you very much for your comment.
The quoted paragraph has been rewritten to show a better connection with previous information.
Comment 3
In the version that I am reviewing there are several parts highlighted in yellow, this is the result of you uploading it like this by mistake, or when I download the file, I see it like this.
Response 3
Thank you very much for your comment. In this case the yellow sections refer to information that it was decided to add. When it was uploaded it should not be in that colour. We apologise for it.
Comment 4
When reading the methodology section, specifically the instruments, I see that within emotional intelligence there are other concepts. I think it is necessary to improve the wording of that part in the introduction so that the reader can easily understand the information.
Response 4
Thank you very much for your suggestion. The variables attention, clarity and emotional repair have been contextualised.
Comment 5
In lines 257-258 they expose numerous investigations, but there is only one investigation that supports it. Improve writing or add more references.
Response 5
Thank you very much for your comment. Two further studies related to the statement you refer to have been added.
Comment 6
Within the limitations of the study, it may be that they have used questionnaires to measure the variables, which may cause that ...
Response 6
Thank you very much for your comment. The limitations related to the questionnaires have been added.
Comment 7
In the final part of the discussion, add information that highlights what is new about your research, and the practical utility of your results.
Response 7
Thank you very much for your comment. Practical applications derived from this study have been added. These are found in the final part of the discussion.
Comment 8
In the conclusions it is necessary to say which of the hypotheses were accepted or rejected.
Response 8
Thank you very much for your comment.
It has been added to the section on conclusions which have been complied with.
Thank you!
Reviewer 2 Report
With regard to manuscript: Study of Physical Activity Practice and Adherence to the Medi-2 terranean Diet on Emotional Intelligence in Elementary School Education Students, for consideration in Children. The paper is worthy of publication, however, I have to be honest in mentioning the absence of more data collected from questionnaire. With the utmost respect, allow me to give you a few suggestions.
· In the introduction, the novelty of the study could be more highlighted. The authors would explain why their findings aggregate the existing knowledge.
· Three-dimensional perspective of emotional intelligence could be better defined to facilitate the reviewer/reader.
· The authors do not explain the reason why they decided to explore the Emotional Intelligence. Why not mood related variables ?
· Describe the numbers o figure 2 in more detail (in terms of interpretation/ meaning).
· Give more details on the place of evaluations, time of day, time in each individual evaluation, training of researcher and other minor things.
· e1, e2 and e3 (figure 1) should be highlighted and explained in analyses. It would be interesting to know a more complete description.
· Give more details on data (mean, median, variation) obtained from Trait Meta Mood Scale, KIDMED Questionnaire, PAQ-C questionnaire.
· It is known (informally) that youngers are resistant to diet changes. This would have any influence on their findings. I would like this point of view to be more in-depth in discussion.
· Didactics would improve with the inclusion of more figures (some scheme drawn by the authors) explaining their findings.
Moderate editing of English language required
Author Response
Dear Reviewer,
With regard to manuscript: Study of Physical Activity Practice and Adherence to the Medi-2 terranean Diet on Emotional Intelligence in Elementary School Education Students, for consideration in Children. The paper is worthy of publication; however, I have to be honest in mentioning the absence of more data collected from questionnaire. With the utmost respect, allow me to give you a few suggestions.
Response: Thank you very much for your comment. The authors believe that through your suggestions, the research will be improved.
Comment 1
In the introduction, the novelty of the study could be more highlighted. The authors would explain why their findings aggregate the existing knowledge.
Response 1
Thank you very much for your comment. What this study brings to the scientific field has been added.
Comment 2
Three-dimensional perspective of emotional intelligence could be better defined to facilitate the reviewer/reader.
Response 2
Thank you very much for your suggestion. The variables attention, clarity and emotional repair have been contextualised.
Comment 3
The authors do not explain the reason why they decided to explore the Emotional Intelligence. Why not mood related variables?
Response 3
Thank you very much for your comment.
According to your comment from line 85 to line 91, the importance of emotional intelligence is justified. The need to study emotional intelligence is defined by its importance. It has been observed that moods are defined by intelligence. It has been found that people who show adequate emotional training show better moods. A new paragraph has been added to justify the above.
Comment 4
Describe the numbers or figure 2 in more detail (in terms of interpretation/ meaning).
Response 4
Thank you very much for your comment.
In this case figure 2 represents a summary of table 1. It symbolises the direction of the effect together with the standardised regression weight. When displaying the standardised regression weight together with the theoretical model, the program rounds off. Figure 3 and 4 are supposed to be supports of table 3 and table 4 respectively. In this case and following your suggestion, the comment has been redrafted.
Comment 5
Give more details on the place of evaluations, time of day, time in each individual evaluation, training of researcher and other minor things.
Response 5
Thank you very much for your comment. Minor details have been added in Procedure section.
Comment 6
e1, e2 and e3 (figure 1) should be highlighted and explained in analyses. It would be interesting to know a more complete description.
Response 6
Thank you very much for your suggestion.
In response to your comment in the Data Analysis section, it has been added to symbolise e1, e2 and e3.
Comment 7
Give more details on data (mean, median, variation) obtained from Trait Meta Mood Scale, KIDMED Questionnaire, PAQ-C questionnaire.
Response 7
Thank you very much for your comment. In this case the authors are unable to provide these data as they have been previously published in other research. Furthermore, according to the research objectives and hypotheses, these data are not related, as the aim is not to describe the levels of emotional intelligence, physical activity and adherence to the Mediterranean diet.
Comment 8
It is known (informally) that youngers are resistant to diet changes. This would have any influence on their findings. I would like this point of view to be more in-depth in discussion.
Response 8
Thank you very much for your comment. In this case we have proceeded to discuss the above paragraph in greater depth.
Comment 9
Didactics would improve with the inclusion of more figures (some scheme drawn by the authors) explaining their findings.
Response 9
Thank you very much for your comment. The authors find your comment very interesting. That is why your suggestion will be implemented in future research.
Thank you!
Reviewer 3 Report
Abstract
Line 16: “Numerous research studies have evidenced that a physical and healthy lifestyle has a beneficial impact on the physical and mental health of young people”… Physical? You talk about physical activity?
Sex! I suggest that you adopted gender.
Introduction
Line 34: “Although this stage shows a high degree of relevance in terms of the acquisition of a healthy lifestyle, it has been observed that the sex of the participants also conditions adherence to a healthy and active lifestyle”…It would very interest explain how and why it happens.
Line 85-89. This paragraph talks about an emotional problematic of this age. Make sense it appears in this part of this section?
Identify independent and dependent variables is hard after read introduction. This section should reorganize.
H2 should reformulate on better way. Furthermore, inclusion of variable sex is unjustified in introduction section.
Emotional Intelligence should be more explored in introduction.
Methods
The measurements used are validated to this age?
Why were used two versions of SPSS?
Structural equation models were performed in SPSS or in another software with connection with SPSS like AMOS?
There is any care about the normality of the data? Mardia coeficient per example.
Figure 1. Model Theoretical – At this point, appears some variables for the first time in this document. Emotional attention; emotional clarity and emotional reparation should be aboard in introduction section as well.
Results
A table with values of CFI, IFI, GFI, TLI will be more readable.
Discussion / Conclusion
Line 220: “Once the research hypotheses and objectives have been answered, this section aims 220 to compare the main findings with those of other similar studies”. Discussion section is more than compare the main findings with other studies.
It will interesting a discussion about the weight “size”, once all them are to low.
Author Response
Dear Reviewer,
Comment 1
Line 16: “Numerous research studies have evidenced that a physical and healthy lifestyle has a beneficial impact on the physical and mental health of young people”… Physical? You talk about physical activity?
Response 1
Thank you very much for your comment.
Physical refers to Physical activity. To avoid misunderstanding, the word activity has been added.
Comment 2
Sex! I suggest that you adopted gender.
Response 2
Thank you for your comment. The word sex has been replaced by gender
Comment 3
Line 34: “Although this stage shows a high degree of relevance in terms of the acquisition of a healthy lifestyle, it has been observed that the sex of the participants also conditions adherence to a healthy and active lifestyle”…It would very interest explain how and why it happens.
Response 3
Thank you for your interest.
The justification for this statement has been added.
Comment 4
Line 85-89. This paragraph talks about an emotional problematic of this age. Make sense it appears in this part of this section?
Response 4
Thank you very much for your question.
The main reason why it was decided to add that paragraph is to highlight the problem between the emotional and academic spheres. It is true that many young people increase their stress and anxiety levels because of academics. This increase in negative emotions is preceded by low emotional competence. That is why it was decided to include this section.
In this case, the authors consider it necessary to leave this justification as it justifies why it has been decided to carry out this study in this population.
Comment 5
Identify independent and dependent variables is hard after read introduction. This section should reorganize.
Response 5
Thank you for your interest. In this case the introduction is structured as follows
Line 20-42: Contextualisation of the target population and problems related to adherence to the Mediterranean diet.
Line 43-61: Contextualisation of the Mediterranean Diet and benefits of an active and healthy lifestyle on the physical and mental area.
Line 62-75: Contextualisation of the practice of physical activity.
Line 76-87: Benefits of physical activity on an emotional level. Definition of emotional intelligence and what elements it is made up of.
Line 88-94: Justification of the choice of the study population.
With regard to the justification of the independent and dependent variables, both hypotheses 1 and 2 make it clear which are the independent variables (Mediterranean Diet Adherence and Physical activity) and which are the dependent variables (Emotional Attention, Emotional Clarity, Emotional Repair).
Comment 6
H2 should reformulate on better way. Furthermore, inclusion of variable sex is unjustified in introduction section.
Response 6
Thank you very much for your suggestion.
H2 has been reworded. Also, the inclusion of gender is defined (Line 33-38).
Comment 7
Emotional Intelligence should be more explored in introduction
Response 7
Thank you very much for your comment. The variable emotional intelligence has been further explored.
Comment 8
The measurements used are validated to this age?
Response 8
Thank you very much for your comment.
The instruments used have been validated for the study population.
Comment 9
Why were used two versions of SPSS?
Structural equation models were performed in SPSS or in another software with connection with SPSS like AMOS?
Response 9
Thank you very much for your comment.
Two different programs from IBM SPSS have been used.
The first was used to analyse the normality of the results (IBM SPSS Statistics). The second was used to carry out structural equation modelling (IBM SPSS Amos).
Comment 10
There is any care about the normality of the data? Mardia coeficient per example.
Response 10
Thank you very much for your comment.
Given the type of analysis you are considering, it is not necessary to study the normality of the data. Following Hair et al. (2010) it is necessary to take into account the (a) sample size, (b) the minimum ratio of cases per variable (c) the ratio of variables per factor.
However, as requested by the authors, we have proceeded to analyse the distribution of the sample.
Comment 11
Figure 1. Model Theoretical – At this point, appears some variables for the first time in this document. Emotional attention; emotional clarity and emotional reparation should be aboard in introduction section as well.
Response 11
Thank you very much for your comment. The variables you cite have been contextualised in the theoretical framework.
Comment 12
A table with values of CFI, IFI, GFI, TLI will be more readable.
Response 12
Thank you very much for your comment.
A table with the requested adjustment indexes has been added.
Comment 13
Line 220: “Once the research hypotheses and objectives have been answered, this section aims 220 to compare the main findings with those of other similar studies”. Discussion section is more than compare the main findings with other studies.
Response 13
Thank you very much for your comment. The introductory sentence to the discussion has been reworded.
Comment 14
It will interesting a discussion about the weight “size”, once all them are to low.
Response 14
Thank you very much for your comment. As can be seen in the discussion when comparing the results according to sex, there is a greater or lesser effect (Lines 257-259; Lines 267-268).
Nevertheless, we consider this suggestion to be very interesting and will be applied in future research.
Thank you!
Round 2
Reviewer 3 Report
Dear authors, thank for your effort to address my comments.
Congratulations.
Please, change the follow point.
Line 193. Please, remove the reference to Kolmogorov Smirnov and justify the normality of date in line that you wrote in your response to my comments about normality.
Author Response
Dear reviewer,
The changes are highlighted in lines 167-169. Thank you for your comment.
Best regards!